# PINK1 alleviates thermal hypersensitivity in a paclitaxel-induced *Drosophila* model of peripheral neuropathy

**Young Yeon Kim**[1,2◉], **Jeong-Hyun Yoon**[1,2◉], **Jee-Hyun Um**[1,2], **Dae Jin Jeong**[1,2], **Dong Jin Shin**[1,2], **Young Bin Hong**[1,2], **Jong Kuk Kim**[1,3], **Dong Hyun Kim**[1,4], **Changsoo Kim**[5], **Chang Geon Chung**[6], **Sung Bae Lee**[7], **Hyongjong Koh**[1,7], **Jeanho Yun**[1,2]*

**1** Peripheral Neuropathy Research Center, Dong-A University, Busan, Republic of Korea, **2** Department of Biochemistry, College of Medicine, Dong-A University, Busan, Republic of Korea, **3** Department of Neurology, College of Medicine, Dong-A University, Busan, Republic of Korea, **4** Department of Medicinal Biotechnology, College of Health Sciences, Dong-A University, Busan, Republic of Korea, **5** Hormone Research Center, School of Biological Sciences and Technology, Chonnam National University, Gwangju, South Korea, **6** Department of Brain & Cognitive Sciences, Daegu Gyeongbuk Institute of Science and Technology, Daegu, Republic of Korea, **7** Department of Pharmacology, College of Medicine, Dong-A University, Busan, Republic of Korea

◉ These authors contributed equally to this work.
* yunj@dau.ac.kr

**Data Availability Statement:** All relevant data are within the manuscript and its Supporting Information files.

## Abstract

Paclitaxel is a representative anticancer drug that induces chemotherapy-induced peripheral neuropathy (CIPN), a common side effect that limits many anticancer chemotherapies. Although PINK1, a key mediator of mitochondrial quality control, has been shown to protect neuronal cells from various toxic treatments, the role of PINK1 in CIPN has not been investigated. Here, we examined the effect of PINK1 expression on CIPN using a recently established paclitaxel-induced peripheral neuropathy model in *Drosophila* larvae. We found that the class IV dendritic arborization (C4da) sensory neuron-specific expression of PINK1 significantly ameliorated the paclitaxel-induced thermal hyperalgesia phenotype. In contrast, knockdown of PINK1 resulted in an increase in thermal hypersensitivity, suggesting a critical role for PINK1 in sensory neuron-mediated thermal nociceptive sensitivity. Interestingly, analysis of the C4da neuron morphology suggests that PINK1 expression alleviates paclitaxel-induced thermal hypersensitivity by means other than preventing alterations in sensory dendrites in C4da neurons. We found that paclitaxel induces mitochondrial dysfunction in C4da neurons and that PINK1 expression suppressed the paclitaxel-induced increase in mitophagy in C4da neurons. These results suggest that PINK1 mitigates paclitaxel-induced sensory dendrite alterations and restores mitochondrial homeostasis in C4da neurons and that improvement in mitochondrial quality control could be a promising strategy for the treatment of CIPN.

**Funding:** This work was supported by National Research Foundation of Korea (NRF) grants funded by the Korean government (2016R1A5A2007009 and 2019R1A2C2003991) to JY. The funders had no role in study design, data collection and analysis, decision to publish, or preparation of the manuscript.

**Competing interests:** The authors declare that there is no conflict of interest.

## Introduction

Chemotherapy-induced peripheral neuropathy (CIPN) is a common side effect that limits many effective anticancer chemotherapies [1]. CIPN is observed in over 68% of patients that receive chemotherapy [2]. Although various symptoms arise depending on the type, amount, and duration of the anticancer drug, typical clinical features of CIPN include sensory nerve abnormalities such as paresthesia, allodynia and hyperalgesia [2]. These symptoms may persist for several months to several years after the completion of chemotherapy in up to 30% of patients, and in severe cases, CIPN adversely affects the quality of life of cancer survivors for the lifetime [2]. However, although CIPN has a negative impact on the quality of life of cancer patients, there is currently no effective and established preventive treatment. Paclitaxel, a microtubule-stabilizing agent, is widely used to treat various solid tumors and is a representative anticancer drug that induces CIPN [3]. The major dose-limiting side effect of paclitaxel treatment is painful peripheral neuropathy, which is predominantly sensory [4, 5].

*Drosophila* models have been shown to be useful for identifying essential genes required for the thermal nociception response [6, 7]. Class IV multidendritic sensory neurons are responsible for sensory nociception and behavioral responses to various noxious stimuli, including mechanical forces and high temperatures, in *Drosophila* larvae [7, 8]. Upon noxious thermal challenge, third instar (L3) larvae show a characteristic corkscrew-like rolling motion that can be easily analyzed [7]. Recent studies that used the *Drosophila* larval CIPN model for paclitaxel-induced peripheral neuropathy have recapitulated CIPN sensory dysfunction [9, 10]. These studies established that paclitaxel treatment induces hypersensitivity to noxious thermal stimulation, that is, thermal hyperalgesia in L3 larvae. More importantly, these studies also revealed that alterations in the sensory dendrites of class IV dendritic arborization (C4da) sensory neurons are associated with paclitaxel-induced peripheral neuropathy [9, 10]. These results proved that the *Drosophila* larva model of paclitaxel-induced peripheral neuropathy is a robust model to investigate the molecular mechanism of CIPN.

Phosphatase and tensin homologue (PTEN)-induced putative kinase 1 (PINK1) is a mitochondrial Ser/Thr kinase and is known as a key regulator of mitochondrial quality control and mitochondria homeostasis [11–13]. PINK1 works as a molecular sensor for mitochondrial damage [11] and controls critical mechanisms for mitochondria quality control including mitochondria fission, fusion, transport, biogenesis and mitophagy [14]. Consistent with the essential role of PINK1 in mitochondrial function, the loss of PINK1 results in mitochondrial dysfunction and hypersensitivity to toxic stress [15, 16]. In contrast, PINK1 protects neuronal cells against various toxic insults including MPTP and, α-synuclein [17, 18]. However, whether PINK1 expression also has a protective effect on CIPN has not been investigated.

In this study, we examined the effect of PINK1 expression on the paclitaxel-induced CIPN model in *Drosophila* larvae. We found that PINK1 expression significantly ameliorated the paclitaxel-induced thermal hyperalgesia phenotype. Our analysis revealed that PINK1 expression suppressed aberrant paclitaxel-induced alterations in sensory dendrite C4da neurons and mitophagy induction.

## Materials and methods

### *Drosophila* strains

The *ppk-GAL4*, *ppk^{1a}-GAL4* and *UAS-CD4-tdTomato (CD4-tdTom)* lines were kindly provided by Y.N. Jan (University of California, San Francisco, CA). The *UAS-mt-Keima* line was generated previously [19]. The *UAS-PINK1* and *PINK1* RNAi lines were generated previously [16]. *w^{1118}* and (*UAS-GFP dsRNA*; BL9331) lines were obtained from the Bloomington Stock

Center (Indiana University, Bloomington, IN). The *UAS-mito-roGFP2-Orp1* line was a gift from Dr. Tobias Dick (German Cancer Research Center, Heidelberg, Germany).

## Larval thermal nociception assays

The larval thermal nociception assays were performed as described previously [6, 9, 10]. Briefly, L3 larvae (120 h after egg laying [AEL]) were rinsed with distilled water and gently placed on a petri dish. After 10 sec acclimation, the larval abdominal A4-A5 segments were touched under a microscope with a custom-built 0.6-mm-wide thermal probe whose temperature was controlled by a microprocessor. The time required to induce the aversive corkscrew-like rolling response was measured as the withdrawal latency up to the 20-sec cut off. The larvae showing no rolling response within 20 sec were considered to have no response. For each thermal nociception assay, at least 50 larvae were analyzed, and the results are presented as the mean values with standard deviation (SD).

## Paclitaxel treatment

Paclitaxel was administered following the feeding regimen described previously [9]. Briefly, twenty virgin female files were mated with fifty male flies for 48–72 h, and the embryos were collected for 2–4 h on grape juice agar plates (15 ml ddH$_2$O, 5 ml grape juice, 0.6 g agar, 1.1 g sucrose, 0.5 ml ethanol, 0.25 ml acetic acid, supplemented with yeast paste (700 mg baker's yeast in 1 ml ddH$_2$O)). The embryos were grown for 72 h to develop into L3 larvae. Larvae were rinsed with distilled water and transferred to freshly made grape juice agar plates containing either 20 μM paclitaxel or 0.2% DMSO. Larvae were grown for another 48 h before performing the thermal nociception assay.

## Measurement of larval size

To measure larval size, images of larvae were captured using a dissection microscope (OLYMPUS MVX10, Olympus Co., Tokyo, Japan) after the thermal nociception assay. The larval area was calculated using ImageJ software (NIH, Bethesda, MD). At least 30 larvae were analyzed per genotype, and the results are presented as the mean values with SD.

## Analysis of images of C4da neuron dendrites

To determine the dendritic structure of C4da neurons at abdominal segment A4 of L3 larvae, images of the fluorescent plasma membrane marker CD4-tdTomato (CD4-tdTom) [20] were obtained using a Zeiss LSM 800 confocal microscope (Carl Zeiss, Oberkochen, Germany). Confocal image stacks of C4da neuron dendrites were converted to maximum intensity projections using Zeiss Zen software. Dendrite length and the number of dendritic branches were analyzed in ImageJ software using the skeleton plugin function (NIH, Bethesda, MD). At least 5 larvae were analyzed per genotype. The dendrite image analysis was repeated three times, and similar results were observed. The results are presented as the mean values with SD.

## *In vivo* measurement of mitochondrial ROS of C4da neurons

For *in vivo* ROS imaging, *ppk>mito-roGFP2-Orp1* L3 larvae were examined with a Zeiss LSM 800 confocal microscope (Carl Zeiss) with a 405-nm (oxidized) or 488-nm (reduced) excitation laser using 520-nm emission as previously described [21]. At least 4–5 larvae were imaged for each group and the 405-nm/ 488-nm fluorescence intensity was obtained using Zeiss Zen software. The results are presented as the mean values with SD.

## Measurement of mitophagy levels

Mitophagy levels were examined using the pH-dependent, fluorescent probe mt-Keima by confocal microscopy as previously described [19, 22]. To acquire the mt-Keima fluorescence images, $ppk^{1a}>mt\text{-}Keima$ larvae were examined with a Zeiss LSM 800 confocal microscope (Carl Zeiss) equipped with a Plan-Apochromat 10×/0.45 M27, Plan-Apochromat 20×/0.8 M27, and c-Apochromat 40×/1.20 W Korr lens. mt-Keima fluorescence was imaged with two sequential excitation lasers (488 nm and 555 nm) using a 595–700 nm emission bandwidth. Mitophagy was quantified based on the analysis of the mt-Keima confocal images using Zeiss Zen software on a pixel-by-pixel basis, as described previously [19, 22]. The mitophagy level (% of mitophagy) was defined as the number of pixels that have a high red/green ratio divided by the total number of pixels. To quantify the mitophagy level in C4da neurons, at least five larvae samples were used for quantification, and the average values were calculated. In all confocal microscopy analyses, all imaging parameters remained constant, and only the gain level was adjusted to avoid the saturation of any pixel. The results are presented as the mean values with SD.

## Statistical analysis

All data are presented as the means ± SDs. Differences between two experimental groups were analyzed using Student's $t$-test. To compare three or more groups, we used a one-way ANOVA with Sidák correction. A $P$-value of $< 0.05$ was considered statistically significant.

## Genotypes

The following genotypes were used: $ppk>w^{1118}$ ($ppk$-GAL4/+); $ppk^{1a} > CD4$-tdTom ($ppk^{1a}$-GAL4/UAS-CD4-tdTomato); $ppk>PINK1$ (UAS-PINK1/+; $ppk$-GAL4/+); $ppk >GFP$ RNAi ($ppk$-GAL4/UAS-GFP RNAi), $ppk>PINK1$ RNAi ($ppk$-GAL4/UAS-PINK1 RNAi); $ppk^{1a}>CD4$-tdTom, PINK1 (UAS-PINK1/+; $ppk^{1a}$-GAL4,UAS-CD4-tdTomato/+); $ppk^{1a}>mt$-Keima ($ppk^{1a}$-GAL4,UAS-mt-Keima/+), $ppk^{1a}>mt$-Keima, PINK1 (UAS-PINK1/+; $ppk^{1a}$-GAL4,UAS-mt-Keima/+). $ppk>mito$-roGFP2-Orp1 ($ppk$-GAL4/UAS-mito-roGFP2-Orp1).

# Results

## Paclitaxel induces a peripheral neuropathy phenotype in *Drosophila* larvae

To study paclitaxel-induced peripheral neuropathy, we adopted a recently established *Drosophila* thermal nociceptive model [6, 9, 10]. We first measured the time required to induce the corkscrew-like rolling withdrawal response after touching the A4-A5 segment region of the third instar (L3) larvae using a heat probe set to different temperatures. Larvae that did not show the rolling response after 20 sec were considered to have no response. As shown in Fig 1A, all larvae showed no response to the 36˚C heat probe, whereas all larvae showed a rapid rolling response to the 46˚C probe (mean withdrawal latency [MWL] = 1.9 sec). We chose 40˚C for the following heat probe assay because the most dynamic thermal nociception response was observed with this heat probe temperature.

Recently, feeding paclitaxel to L3 larvae was shown to induce hypersensitivity to thermal nociception and dendritic structure alteration of C4da sensory neurons [9, 10]. Similar to these studies, we treated L3 larvae with 20 µM paclitaxel for 48 h and performed the thermal nociception assay (Fig 1B). Consistent with previous studies [9, 10], paclitaxel (20 µM) treatment was sufficient to induce significant hypersensitivity to noxious heat (40˚C) (Fig 1C). MWL was reduced to approximately 58% (from 5.88 sec to 3.41 sec) upon paclitaxel treatment. Next, we compared terminal dendrite morphology and branch points of the control C4da

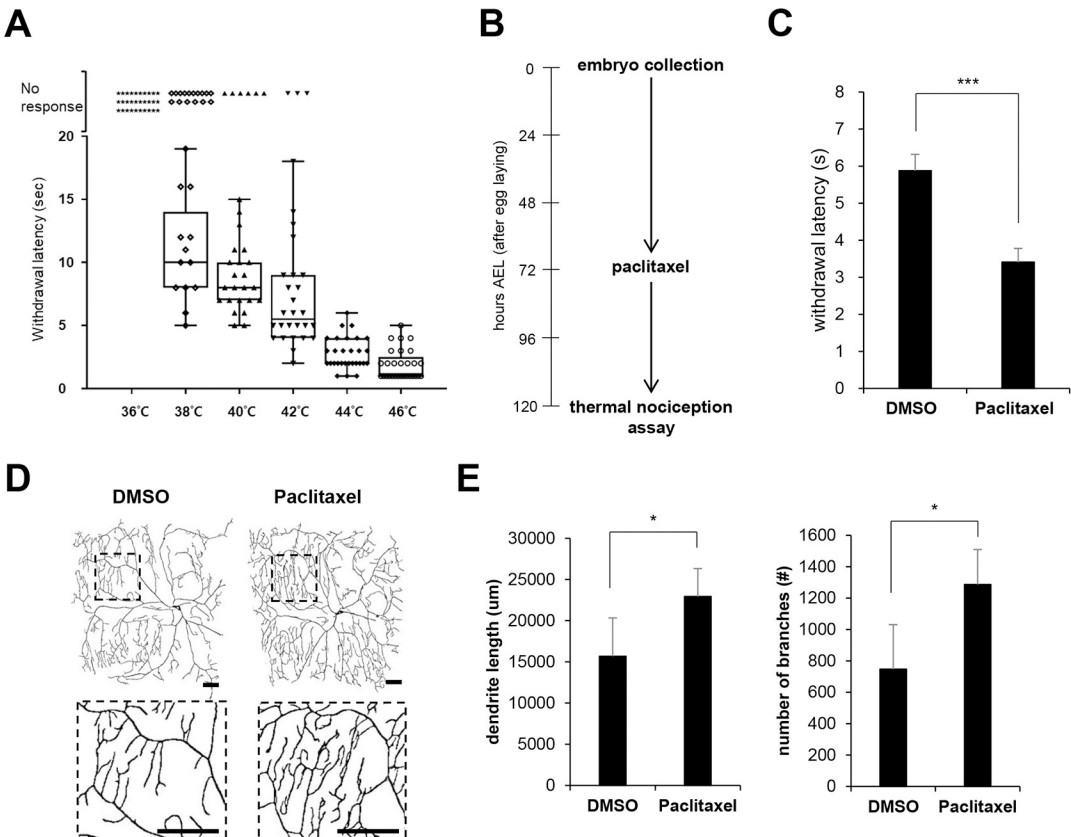

**Fig 1. Paclitaxel treatment induces a heat-hyperalgesia phenotype in *Drosophila* larvae.** (A) Thermal nociceptive response of L3 larvae ($ppk>w^{1118}$) to heat probes at different temperatures. Each data point represents the withdrawal latency of an individual larva. The absence of an aversive rolling response within 20 sec was considered no response. n = 50 larvae were tested at each temperature. (B) Experimental design for paclitaxel treatment and the thermal nociception assay. The early L3 larvae were transferred to medium containing either DMSO vehicle or paclitaxel (20 μM) at 72 h AEL. The larvae were treated for 48 h, and the thermal nociception response was observed at 120 AEL. (C) Thermal nociceptive withdrawal upon 40°C stimulation was assessed after either paclitaxel (20 μM) or DMSO treatment as in (B) (n = 50 per sample). (D) Representative images of C4da neurons at abdominal segment A4 in L3 larvae ($ppk^{1a} > CD4\text{-}tdTom$) expressing the plasma membrane marker CD4-tdTom after 48 h exposure to either vehicle (DMSO) or 20 μM paclitaxel according to the paclitaxel treatment regimen in (B). The boxed regions are shown enlarged in the bottom panel. Scale bars, 50 μm. (E) Quantification of the length of dendrites and the number of dendritic branch points of C4da neurons. n = 5 per sample. The results are presented as the mean values, and the error bars represent the SD. *$P < 0.05$; ***$P < 0.001$ as determined by Student's t-test.

neurons after paclitaxel treatment by visualizing C4da neuron dendrites using the plasma membrane marker CD4-tdTomato, which efficiently labels terminal dendrite branches [20]. In addition to the increased thermal nociception in the heat probe assay, paclitaxel treatment has been shown to lead to alterations in sensory neuron structure [9, 10]. Consistent with these reports, we observed a significant increase in dendrite length and in the number of dendrite branch points (Fig 1D and 1E). These results indicate that our paclitaxel feeding regimen is feasible to develop a peripheral neuropathy phenotype in *Drosophila* larvae.

## Ectopic expression of PINK1 rescues the thermal hypersensitivity phenotype induced by paclitaxel treatment

To examine the effect of PINK1 on paclitaxel-induced thermal sensory hypersensitivity, we expressed PINK1 specifically within C4da neurons using the *ppk-GAL4* driver [20] and examined thermal nociception upon paclitaxel treatment using a *ppk-GAL4* as a control. We found

that PINK1 overexpression significantly reduced thermal sensitivity (Fig 2A). Larvae expressing PINK1 exhibited approximately 13% to 48% increased withdrawal latency with a different heat probe temperature, suggesting that PINK1 is implicated in thermal nociception. Interestingly, we observed that PINK1 overexpression in C4da neurons significantly suppressed the sensitivity to heat upon paclitaxel treatment. While paclitaxel reduced the MWL by 2.41 sec (from 5.88 sec to 3.47 sec) in the control larvae (*ppk>w^1118*), the MWL was decreased by0.99 sec (from 8.08 sec to 7.09 sec) upon paclitaxel treatment in the larvae expressing PINK1 (*ppk>PINK1*) (Fig 2B). In terms of the relative changes, paclitaxel reduced the MWL by 42% in control larvae, while the MWL was decreased by only 12% in larvae expressing PINK1 upon

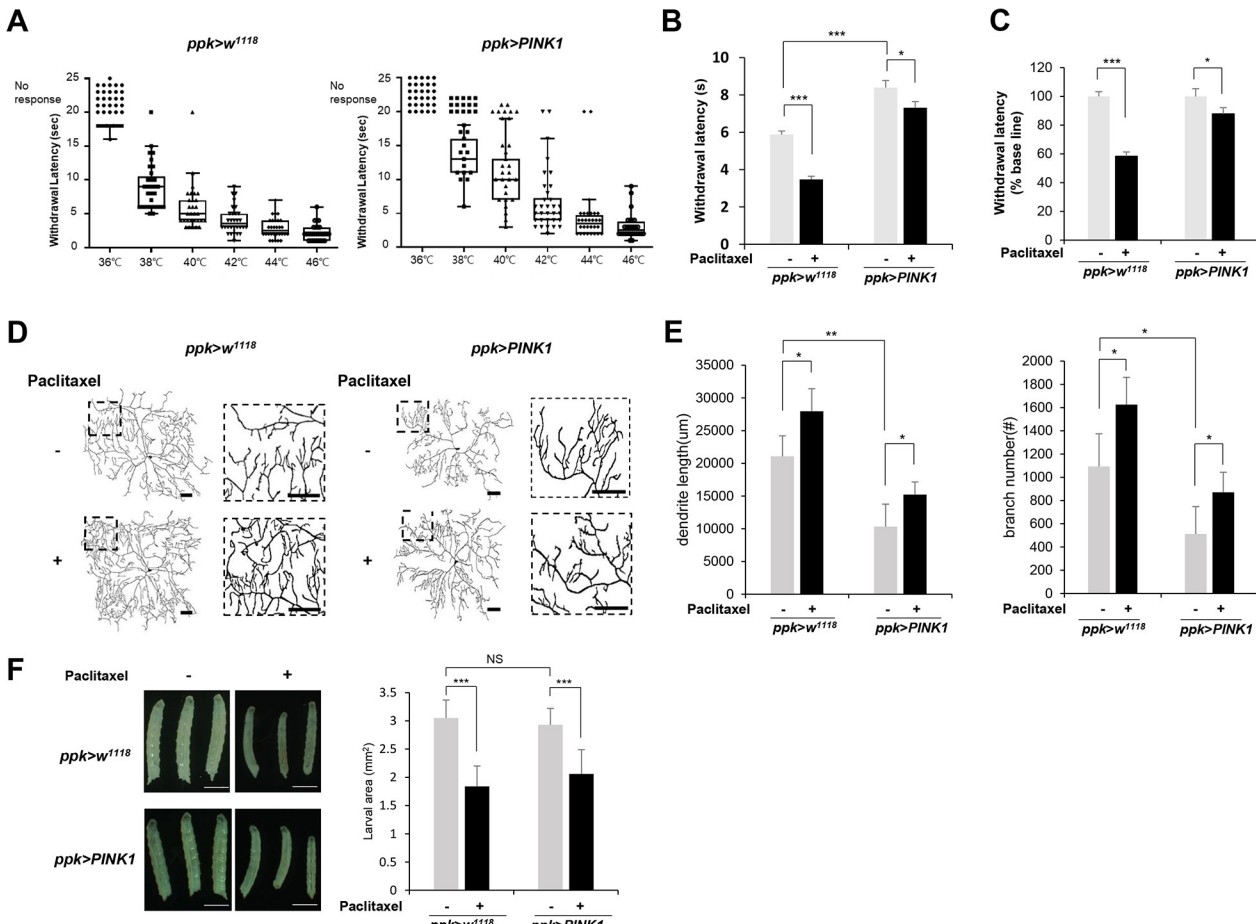

**Fig 2. PINK1 mitigates the heat-hyperalgesia phenotype induced by paclitaxel treatment.** (A) The thermal nociceptive response of *ppk-GAL4* control L3 larvae (*ppk>w^1118*) and larvae expressing PINK1 in C4da sensory neuron (*ppk>PINK1*) to heat probes at different temperatures. Each data point represents the withdrawal latency of an individual larva. The absence of an aversive rolling response within 20 sec was considered no response. n = 50 larvae were tested at each temperature. (B) Thermal nociceptive withdrawal of *ppk>w^1118* and *ppk>PINK1* larvae from heat probe (40˚C) after 48 h of exposure to either vehicle (DMSO) or 20 μM paclitaxel (n = 50 per sample). (C) The relative withdrawal latency of *ppk>w^1118* and *ppk>PINK1* larvae upon paclitaxel treatment was calculated according to that of the vehicle sample of each genotype, which was considered to be 100%. (D) Representative images of C4da neurons at abdominal segment A4 of control L3 larvae (*ppk^1a > CD4-tdTom*) and L3 larvae expressing PINK1 (*ppk^1a > CD4-tdTom,PINK1*). Larvae were treated with paclitaxel (20 μM) for 48 h. Right images are enlargements of the boxed regions in the left images. Scale bars, 50 μm. (E) Quantification of the length of dendrites (*left*) and the number of dendrite branch points (*right*) in C4da neurons. n = 5 per sample. (F) Representative images of larvae from each genotype at 120 AEL treated with either DMSO or paclitaxel for 48 h (*left*). The larval areas calculated by the length multiplied by the width of each larva of each genotype after either DMSO or paclitaxel treatment were plotted (n≧50 per sample) (*right*). Scale bars, 1 mm. The results are presented as the mean values, and the error bars represent the SD. Significance was determined by one-way ANOVA with Sidák correction. $^*P < 0.05$; $^{**}P < 0.01$; $^{***}P < 0.001$. NS; not significant.

paclitaxel treatment (Fig 2C). These results suggest that PINK1 expression significantly alleviated paclitaxel-induced thermal hypersensitivity in a thermal nociception assay.

We then assessed the dendritic arborization of C4da neurons, which is known to be associated with the thermal nociception [8]. We observed significant decreases in dendrite length and the number of branch points in larvae expressing PINK1 (Fig 2D and 2E). Ectopic expression of PINK1 reduced the dendrite length and the number of branch points of C4da neurons by 51% and 53%, respectively, compared to those of the control larvae, suggesting that the reduced thermal sensitivity upon PINK1 expression may be due to decreased dendritic arborization of C4da neurons. Because PINK1 expression suppressed paclitaxel-induced thermal hypersensitivity, we expected that dendritic arborization would increase less in larvae expressing PINK1 than in control larvae. However, paclitaxel treatment further increased dendritic arborization in larvae expressing PINK1 compared with control larvae (Fig 2D and 2E). Quantitative analysis of multiple C4da neurons (n ≧ 5) revealed that the dendrite length was increased upon paclitaxel treatment by 32% in control larvae and by 42% in larvae expressing PINK1 (Fig 2E). The number of dendrite branch points was increased upon paclitaxel treatment by 49% and 70% in the control larvae and in the larvae expressing PINK1 respectively (Fig 2E). These results suggest that suppressing alteration of dendrite arborization may not be the only mechanism through which PINK1 alleviates the thermal hypersensitivity induced by paclitaxel treatment.

A previous study has shown that paclitaxel treatment in L3 larvae slightly inhibited larval growth [9]. The size of larvae expressing PINK1 in C4da neurons was comparable to that of the control larvae (Fig 2F), indicating that PINK1 overexpression in C4da neurons does not interfere with larval growth. In addition, paclitaxel induced a similar level of inhibition of larval growth in both control and PINK1-expressing larvae (Fig 2F). The size of the larvae was reduced upon paclitaxel treatment in control larvae ($ppk>w^{1118}$) and in larvae expressing PINK1 ($ppk>PINK1$) to 58% and 67%, respectively. These results also suggest that PINK1 expression does not significantly interfere with the inhibitory effect of paclitaxel on larval growth, although the basal MWL was increased.

## PINK1 knockdown induces thermal hypersensitivity

To further test the role of PINK1 in thermal nociception, we next examined the effect of specific knockdown of PINK1 expression in C4da neurons. Whereas ectopic PINK1 expression reduced thermal sensitivity in the heat probe assay, C4da neuron-specific knockdown of PINK1 using the *ppk-GAL4* driver resulted in significant sensitization of L3 larvae to different heat probe temperatures in the thermal nociception assay (Fig 3A). PINK1 knockdown decreased the withdrawal latency to the 40°C heat probe to 41% of the control level (Fig 3B), confirming that PINK1 plays an important role in thermal nociception.

To understand the role of PINK1 in the sensory dendrites of C4da neurons, we also examined the dendritic structure of C4da neurons upon C4da neuron-specific knockdown of PINK1. Interestingly, we observed no change in the dendritic structure of C4da neurons upon PINK1 knockdown (Fig 3C). The quantitative analysis of multiple C4da neurons (n ≧ 5) also showed that both dendrite length and the number of dendrite branches (Fig 3D) were not significantly changed upon PINK1 knockdown. These results indicated that knockdown of PINK1 in C4da neurons induces an increase in thermal nociception without changes in the dendritic structure. We also found that the size of larvae was not significantly changed upon PINK1 knockdown (Fig 3E), confirming that PINK1 has no effect on larval growth. These results suggest that PINK1 may separately regulate thermal nociception and dendrite structure through independent mechanisms.

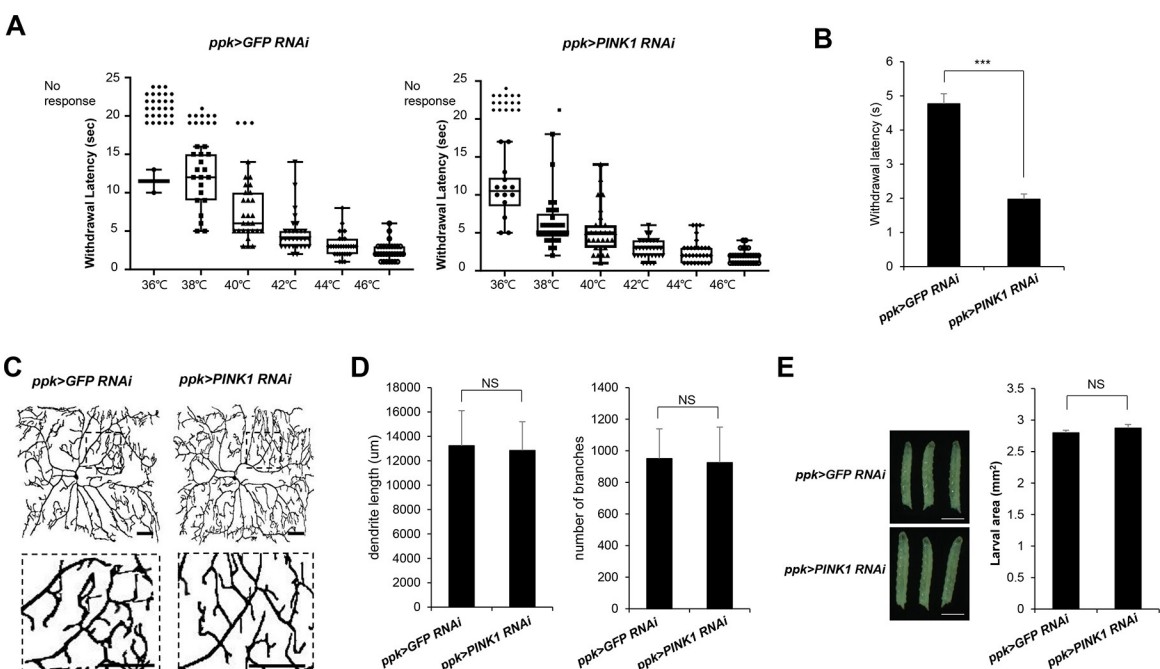

**Fig 3. Effect of PINK1 knockdown on thermal nociception in *Drosophila* larvae.** (A) The thermal nociceptive profiles of L3 *ppk>GFP RNAi* and *ppk>PINK1 RNAi* larvae in response to heat probes at different temperatures. Each data point represents the withdrawal latency of an individual larva. The absence of an aversive rolling response within 20 sec was considered no response. n = 50 larvae were tested at each temperature. (B) Thermal nociceptive withdrawal of *ppk>GFP RNAi* and *ppk>PINK1 RNAi* to a heat probe (40°C) was examined in the L3 larvae stage (120 h AEL). (n = 50 per sample). (C) Representative images of C4da neurons at abdominal segment A4 of control L3 larvae (*ppk^{1a} > CD4-tdTom, GFP RNAi*) and L3 larvae expressing PINK1 (*ppk^{1a} > CD4-tdTom, PINK1 RNAi*). Larvae were treated with paclitaxel (20 μM) for 48 h. Right images are enlargements of the boxed regions in the left images. The boxed regions are shown enlarged in the bottom panel. Scale bars, 50 μm. (D) Quantification of the length of the dendrites (*left*) and the number of dendrite branch points (*right*) in C4da neurons. n = 5 per sample. Scale bars, 1 mm. (E) Representative images of larvae from each genotype at 120 h AEL (*left*). The larval areas were calculated by the length multiplied by the width of each larva of each genotype (n = 50 per sample) (*right*). The results are presented as the mean values, and the error bars represent the SD. ***$P < 0.001$ as determined by Student's t-test. NS; not significant.

## PINK1 reduced paclitaxel-induced increases in mitophagy levels in C4da sensory neurons

Recent studies showed that paclitaxel induces mitochondrial dysfunction and this mitochondrial dysfunction could be a possible cause of paclitaxel-induced sensory peripheral neuropathy [23, 24]. To understand how PINK1 mitigates paclitaxel-induced thermal hypersensitivity, we next examined whether paclitaxel induces mitochondrial dysfunction in our paclitaxel-induced peripheral neuropathy model. It has been shown that mitochondrial ROS levels correlate with mitochondrial dysfunction in *Drosophila* [25]. To examine mitochondrial dysfunction upon paclitaxel treatment, we measured the mitochondrial ROS levels of C4da neurons by expressing the *in vivo* mitochondrial $H_2O_2$ probe mito-roGFP2-Orp1 [21] specifically within C4da neurons. As shown in Fig 4, we observed that the mitochondrial ROS level was significantly increased upon paclitaxel treatment in C4da neurons. These results suggest that paclitaxel induces mitochondrial dysfunction in C4da neurons.

Mitochondrial dysfunction is known to induce mitophagy, a selective process for the degradation of damaged or dysfunctional mitochondria [26, 27]. We also previously showed that various insults inducing mitochondrial dysfunction such as the loss of the mitochondrial polymerase γ (POLG) gene, hypoxia, or rotenone treatment results in increased mitophagy [19, 22]. Then, we next examined the level of mitophagy in C4da sensory neurons upon paclitaxel

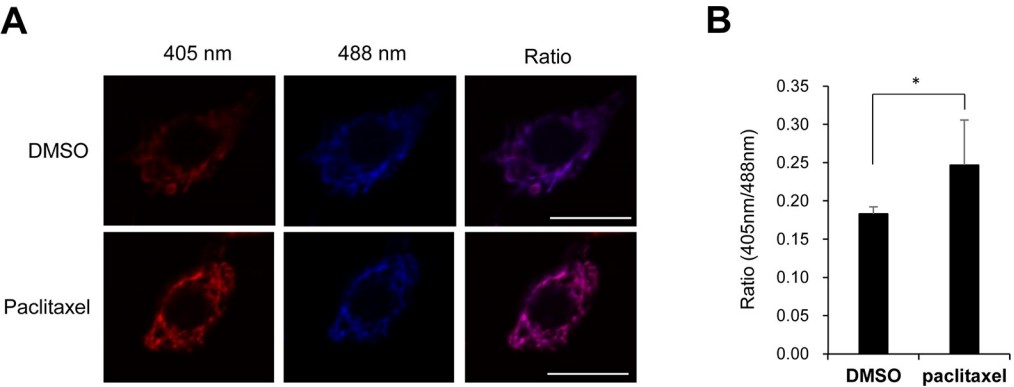

**Fig 4. Increased mitochondrial ROS upon paclitaxel treatment in C4da neurons.** (A) Representative fluorescence images of C4da sensory neurons at abdominal segment A4 in L3 larvae expressing the *in vivo* mitochondrial $H_2O_2$ probe mito-roGFP2-Orp1 (*ppk>mito-roGFP2-Orp1*) with either DMSO or paclitaxel (20 μM) for 48 h. Scale bars, 10 μm. (B) Quantitative analysis of the mitochondrial ROS levels of the C4da sensory neurons in each group (n = 4 or 5 per group). The results are presented as the mean values, and the error bars represent the SD. *$P < 0.05$ as determined by Student's t-test.

treatment. To measure the mitophagy level in C4da sensory neurons, we specifically expressed a pH-dependent fluorescent protein probe, mitochondria-targeted Keima (mt-Keima) [19, 22], using the *ppk-GAL4* driver and quantitatively measured mitophagy activity as described recently [19]. Red puncta, an indicator of mitophagic mitochondria, were increased upon paclitaxel treatment (Fig 5A). The quantitative analysis of multiple C4da neurons revealed that the mitophagy level was increased by approximately 2.5-fold upon paclitaxel treatment (Fig 5B), suggesting paclitaxel induces mitochondrial dysfunction in C4da sensory neurons. PINK1 has

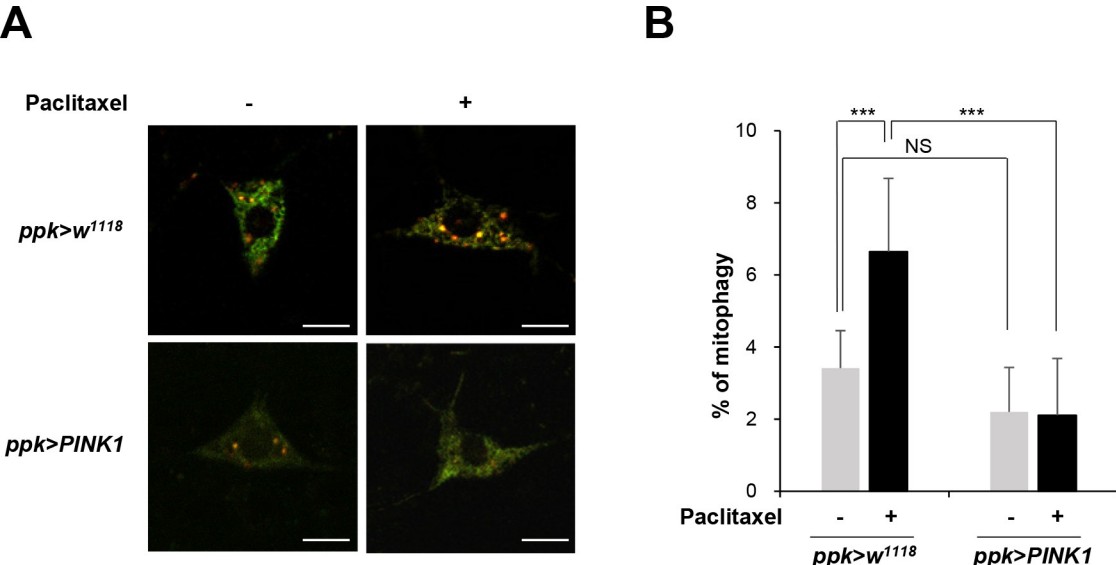

**Fig 5. PINK1 restores mitochondrial homeostasis in paclitaxel-treated sensory neurons.** (A) Representative mt-Keima fluorescence images of C4da sensory neurons at abdominal segment A4 in control L3 larvae (*ppk^1a > mt-Keima*) and L3 larvae expressing PINK1 (*ppk^1a > mt-Keima, PINK1*) treated with either DMSO or paclitaxel (20 μM) for 48 h. Scale bars, 10 μm. (B) Quantitative analysis of the mitophagy of C4da sensory neurons in each group (n = 10 per group). The results are presented as the mean values, and the error bars represent the SD. Significance was determined by one-way ANOVA with Sidák correction. **$P < 0.01$.

been shown to reduce the aberrant increase in mitophagy in response to toxic treatment in SH-SY5Y cells [28]. Consistently, the paclitaxel-induced increase in mitophagy in C4da sensory neurons was significantly suppressed by PINK1 expression (Fig 5A and 5B). We did not observe significant differences in the levels of basal mitophagy in the C4da neurons after either overexpression or knockdown of PINK1 (Fig 5A and S1 Fig), suggesting the PINK1 function is dispensable for basal mitophagy as reported by previous studies [19, 29]. Together, these results suggest that PINK1 expression reduced mitochondrial dysfunction and restored mitochondria homeostasis in the paclitaxel-induced CIPN model in *Drosophila* larvae.

## Discussion

In this study, we report for the first time the neuroprotective function of PINK1 in a paclitaxel-induced peripheral neuropathy model. First, we found that the overexpression of PINK1 in C4da sensory neurons significantly ameliorates paclitaxel-induced thermal hypersensitivity in *Drosophila* larvae. The critical role of PINK1 in sensory nociception was further confirmed by an increase in thermal hypersensitivity upon the knockdown of PINK1 in C4da neurons.

PINK1 has been shown to protect neuronal cells from various toxic reagents, such as 1-methyl-4-phenyl-1,2,3,6-tetrahydropyridine (MPTP) [30], staurosporin [31], propofol [32], chlorpyrifos [33], and thapsigargin [34]. PINK1 has also shown protective effects in various neuronal disease models, including an α-synuclein-induced Parkinson's disease model [17, 18, 35, 36], a Huntington's disease model [37], and an Alzheimer's disease model [38]. These studies showed that PINK1 plays critical prosurvival and antiapoptotic functions in neuronal cells. Interestingly, we observed that the paclitaxel-induced larval growth inhibition was not affected by PINK1 expression, suggesting that PINK1 ameliorates the paclitaxel-induced hyperalgesia phenotype through methods other than the inhibition of cell death.

*Drosophila* terminal sensory dendrites have dynamic plasticity, showing both retraction and extension events [39] similar to mammalian skin sensory axons [40]. Brazill et al. recently revealed that paclitaxel treatment increases the stability of terminal dendrites and inhibits terminal branch retraction, leading to increased terminal dendrite density [9]. In the present study, we observed that PINK1 expression induces changes in the dendrite density of C4da sensory neurons, suggesting that PINK1 modulates the dynamic plasticity of sensory neurons. Recent studies have also shown that PINK1 regulates dendrite morphogenesis and neuronal plasticity [41, 42]. Thus, the results from our group and others suggest that PINK1 is an important regulator of dendrite plasticity. Interestingly, knockdown of PINK1 in C4da neurons resulted in increased thermal nociception without changes in dendrite structure. Therefore, our results from ectopic expression of PINK1 and knockdown experiments suggest that PINK1 regulates thermal nociception through a different mechanism than the controlling dendrite structure of C4da neurons.

Beyond its antimicrotubule effects, it has long been known that paclitaxel also acts on mitochondria. [43]. Previous *in vivo* studies also indicated that mitochondrial dysfunction is associated with paclitaxel-induced peripheral neuropathy. Xiao et al and Zheng et al revealed that paclitaxel treatment resulted in mitochondrial dysfunction in a rat model, as evidenced by reduced mitochondrial respiration and swollen and vacuolated mitochondria in sensory neurons [23, 24]. Paclitaxel caused an increase in mitochondrial ROS, increased mitochondrial volume and dysregulated intracellular $Ca^{2+}$ [44, 45]. These studies suggest that the neuronal damage induced by paclitaxel treatment is closely associated with mitochondrial dysfunction. Previous studies have shown that the protective effect of PINK1 relies on its role in ameliorating mitochondrial dysfunction [37, 38]. In contrast, the loss of PINK1 resulted in severe mitochondrial dysfunction in a *Drosophila* model [16], a cell line [28] and a mouse model [46].

These results indicate that PINK1 plays a critical role in maintaining mitochondrial homeostasis.

PINK1 may control thermal nociception by reducing mitochondrial dysfunction in C4da neurons. Dagda et al. previously showed that the overexpression of PINK1 suppressed toxin-induced mitophagy, while the knockdown of PINK1 induced mitochondrial ROS and morphological changes [28]. Consistent with this study, we also observed that PINK1 expression suppressed the paclitaxel-induced increase of mitophagy in C4da sensory neurons. The restoration of mitophagy levels in C4da neurons to normal levels upon PINK1 expression suggests that mitochondrial homeostasis is restored by PINK1. Recent studies have shown that loss of PINK1 had no significant effect on the base mitophagy levels in different tissues such as muscle and DA neurons [19, 29, 47]. In the present study, we also observed that the mitophagy level of C4da neurons in L3 larvae was not significantly changed by either overexpression or knockdown of PINK1. Given the significant increase in thermal sensitivity upon PINK1 knockdown, the unchanged mitophagy activity suggests that PINK1 may restore mitochondrial homeostasis through mechanisms other than mitophagy. Previous studies have indicated that PINK1 controls mitochondrial quality through additional ways such as regulating complex I activity and mitochondrial transport [13, 48]. A body of studies even suggest that PINK1 suppresses the mitophagy activity in certain cellular contexts in a direct or indirect manner [48]. Therefore, the exact mechanism by which PINK1 ameliorates paclitaxel-induced mitochondrial dysfunction in C4da neurons should be determined by further studies. Further studies investigating how PINK1 restores mitochondrial function in paclitaxel-treated sensory neurons and the functional relationship between mitochondrial function and thermal nociception upon paclitaxel treatment will provide valuable information about the molecular mechanism responsible for paclitaxel-induced peripheral neuropathy. In addition, because PINK1 overexpression induces alteration of the dendrite structure of C4da neuron as well as thermal nociception, pharmacological transient activation of PINK1 may be an efficient strategy to control paclitaxel-induced thermal hypersensitivity while mitigating possible side effects from alteration of the dendrite structure of sensory neurons.

In conclusion, our study provides evidence that the ectopic expression of PINK1 ameliorates the thermal hypersensitive phenotype of paclitaxel-induced peripheral neuropathy. Although the precise mechanism by which PINK1 expression suppresses paclitaxel-induced mitochondrial dysfunction in sensory neurons remains to be explored, our results highlight PINK1 as a potential target for the treatment of paclitaxel-induced peripheral neuropathy.

## Supporting information

**S1 Fig. Effect of PINK1 knockdown on the mitophagy level in C4da neuron of L3 larvae.** Quantitative analysis of the mitophagy of C4da sensory neurons at abdominal segment A4 in L3 *ppk>GFP RNAi* and *ppk>PINK1 RNAi* larvae (n = 5 per group). The results are presented as the mean values, and the error bars represent the SD. Significance was determined by Student's t-test. NS; not significant.
(DOCX)

## Author Contributions

**Conceptualization:** Young Yeon Kim, Jeong-Hyun Yoon, Jeanho Yun.

**Formal analysis:** Young Yeon Kim, Jeong-Hyun Yoon, Jee-Hyun Um.

**Investigation:** Young Yeon Kim, Jeong-Hyun Yoon, Jee-Hyun Um, Dae Jin Jeong, Dong Jin Shin.

**Methodology:** Young Yeon Kim, Jeong-Hyun Yoon, Changsoo Kim, Chang Geon Chung, Sung Bae Lee, Hyongjong Koh.

**Resources:** Young Bin Hong, Jong Kuk Kim, Dong Hyun Kim, Sung Bae Lee, Hyongjong Koh.

**Supervision:** Young Bin Hong, Jong Kuk Kim, Dong Hyun Kim, Jeanho Yun.

**Validation:** Changsoo Kim, Sung Bae Lee, Jeanho Yun.

**Writing – original draft:** Young Yeon Kim, Jeong-Hyun Yoon, Jeanho Yun.

**Writing – review & editing:** Young Yeon Kim, Jee-Hyun Um, Hyongjong Koh, Jeanho Yun.

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
