## [Decision Letter · Decision Letter 0]

9 Jan 2020

PONE-D-19-31363

PINK1 alleviates thermal hypersensitivity in a paclitaxel-induced Drosophila model of peripheral neuropathy

PLOS ONE

Dear Prof Yun,

Thank you for submitting your manuscript to PLOS ONE. After careful consideration, we feel that it has merit but does not fully meet PLOS ONE’s publication criteria as it currently stands. Therefore, we invite you to submit a revised version of the manuscript that addresses the points raised during the review process.

We would appreciate receiving your revised manuscript by Feb 23 2020 11:59PM. To enhance the reproducibility of your results, we recommend that if applicable you deposit your laboratory protocols in protocols.io, where a protocol can be assigned its own identifier (DOI) such that it can be cited independently in the future. For instructions see: http://journals.plos.org/plosone/s/submission-guidelines#loc-laboratory-protocols

We look forward to receiving your revised manuscript.

Kind regards,

David Chau

Academic Editor

PLOS ONE

Journal Requirements:

Reviewers' comments:

Reviewer's Responses to Questions

**Comments to the Author**

1. Is the manuscript technically sound, and do the data support the conclusions?

Reviewer #1: Yes

Reviewer #2: Yes

2. Has the statistical analysis been performed appropriately and rigorously? 

Reviewer #1: Yes

Reviewer #2: Yes

3. Have the authors made all data underlying the findings in their manuscript fully available?

Reviewer #1: Yes

Reviewer #2: Yes

4. Is the manuscript presented in an intelligible fashion and written in standard English?

Reviewer #1: Yes

Reviewer #2: Yes

5. Review Comments to the Author

Reviewer #1: This study focuses on the role of PINK1 in chemotherapy induced pain. The study is clearly presented and executed reasonably. The results are interesting to both the pain field and probably also the neurodegeneration community.

I have only minor comments:

1. Please show temperature dose response profiles similar to Fig 1 a for PPK-GFP vs PPK-PINK1 overexpression or control vs RNAi so we understand the baseline difference in nociception.

2. PDF figures are of low quality please improve.

Reviewer #2: Kim et al, in this manuscript found that overexpressing pink1 rescue paclitaxel-induced Drosophila model of peripheral neuropathy. This CIPN) model was previously established by other labs. Overall, the phenotype sounds interesting, while the underlying molecular mechanism is elusive. The authors need address the following questions for further consideration.

Point 1, data showed in Figure1 is quite similar with a previous publication in DMM PMID: 6031360, where the assay was developed. They just verified this assay and no point to present as a main figure.

Point 2. Fig2 and Fig4 showed that overexpressing pink1 can rescue branch numbers, dendrite length and relative withdrawal for CIPN, since they addressed the same point, they could be merged in a single figure.

Point 3. Phenotypical analysis such as branch numbers, dendrite length should also be done for pink1 RNAi in Fig 3. Also, the phenotype when knocking down pink1 in this CIPN model need to be addressed.

Point 4. In Fig4, the authors showed that mitophagy defects in this CIPN model can also be restored by overexpressing pink1.

This correlation study failed to provide in-depth information about the role of pink1 in this context. A couple of experiments may be helpful to address this question:

For instance, is parkin or other autophagic genes involved?

Is mitochondrial function normal?

Mitochondrial membrane potential or ROS level can be easily tested.

In addition, since paclitaxel targets microtube, mitochondrial transport might also be affected. They should test it.

Point 5. Loss of Pink1/parkin cause neurodegeneration, whether pink1 cause C4da neurons death was not tested in this context.

6. PLOS authors have the option to publish the peer review history of their article (what does this mean?). If published, this will include your full peer review and any attached files.

Reviewer #1: No

Reviewer #2: No

---

## [Author Response · Author response to Decision Letter 0]

11 Mar 2020

March 9, 2020

Dear Editor,

Thank you very much for reviewing our manuscript entitled “PINK1 alleviates thermal hypersensitivity in a paclitaxel-induced Drosophila model of peripheral neuropathy” (PONE-D-19-31363). We greatly appreciate the reviewers’ valuable comments and suggestions. We have modified the manuscript extensively and have included additional data based on the reviewers’ suggestions, which we believe have strengthened the manuscript.

We hope that we have fully addressed the concerns of each reviewer and that the revised manuscript now meets the standards for publication in PLoS One.

Each concern raised by the reviewers is carefully addressed point-by-point below.

Reviewer #1: 

This study focuses on the role of PINK1 in chemotherapy induced pain. The study is clearly presented and executed reasonably. The results are interesting to both the pain field and probably also the neurodegeneration community.

I have only minor comments:

1. Please show temperature dose response profiles similar to Fig 1 a for PPK-GFP vs PPK-PINK1 overexpression or control vs RNAi so we understand the baseline difference in nociception.

Response: We appreciate the reviewer for their positive comments. According to the reviewer’s suggestion, we measured the thermal nociceptive response of ppk>w1118 and ppk>PINK1 L3 larvae to heat probes at different temperatures. We also measured the thermal nociceptive response of ppk>GFP RNAi and ppk>PINK1 RNAi L3 larvae. 

We have added these data as Supplementary Fig. S1 and Fig. S2, respectively, and have modified the Results section accordingly.

2. PDF figures are of low quality please improve.

Response: The low resolution figures in the PDF file are probably caused by the conversion of the original images to the PDF files. All of the submitted figure images were of 300 dpi resolution. In this revision, we have prepared all the original figure images carefully according to the figure preparation guidelines of PLoS One. 

Reviewer #2: 

Kim et al, in this manuscript found that overexpressing pink1 rescue paclitaxel-induced Drosophila model of peripheral neuropathy. This CIPN) model was previously established by other labs. Overall, the phenotype sounds interesting, while the underlying molecular mechanism is elusive. The authors need address the following questions for further consideration.

Point 1, data showed in Figure1 is quite similar with a previous publication in DMM PMID: 6031360, where the assay was developed. They just verified this assay and no point to present as a main figure.

Response: As we have described in the Results section, we established a paclitaxel-induced peripheral neuropathy model and phenotype analysis system referring to several recent studies. Although we adopted the paclitaxel treatment paradigm from Zhai’s groups study (Brazil J.M. et al, 2018, Dis Model Mech, doi: 10.1242/dmm.032938), we established the heat probe assay and dendrite structure analysis assays using different system such as different custom-built thermal probes, different fluorescent membrane markers (CD4-tdTomato), and a dendrite analysis system. We believe that verifying the paclitaxel-induced peripheral neuropathy model is an important step for our study and thus, we want keep Figure 1 as a main figure in our manuscript.

Point 2. Fig2 and Fig4 showed that overexpressing pink1 can rescue branch numbers, dendrite length and relative withdrawal for CIPN, since they addressed the same point, they could be merged in a single figure.

Response: According to the reviewer’s suggestion, we have combined previous Figure 2 and Figure 4 and corrected the manuscript accordingly.

Point 3. Phenotypical analysis such as branch numbers, dendrite length should also be done for pink1 RNAi in Fig 3. Also, the phenotype when knocking down pink1 in this CIPN model need to be addressed.

Response: According to the reviewer’s suggestion, we have examined the dendrite structure of C4da neurons upon PINK1 knockdown. We found that that both the dendrite length and the number of dendrite branches were not significantly changed upon PINK1 knockdown. These results suggest that knockdown of PINK1 has no effect on the dendrite structure of C4da neurons as well as larval growth.

We have added these data to Figure 3(C, D) and have modified the Results section.

Point 4. In Fig4, the authors showed that mitophagy defects in this CIPN model can also be restored by overexpressing pink1.

This correlation study failed to provide in-depth information about the role of pink1 in this context. A couple of experiments may be helpful to address this question:

For instance, is parkin or other autophagic genes involved? Is mitochondrial function normal?

Mitochondrial membrane potential or ROS level can be easily tested.

In addition, since paclitaxel targets microtube, mitochondrial transport might also be affected. They should test it.

Response: We agree with the reviewer that the analysis of the involvement of Parkin or other autophagy genes would provide valuable information for understanding the role of PINK1. Regrettably, we were not able to perform an additional genetic study because the generation of the Drosophila line is not possible within the limited time frame of this revision. Mitochondrial transport was also not examined during this revision due to a lack of a system for the analysis. 

During this revision, we further confirmed paclitaxel-induced mitochondrial dysfunction. Because the analysis of C4da neuron-specific changes in mitochondrial membrane potential and mitochondrial ROS is not possible using conventional fluorescence dyes such as TMRM, TMRE, or mitoSOX, we adopted the mitochondrial H2O2 probe mito-roGFP2-Orp1 (Albrecht, S.C. et al, 2011, Cell Metab, doi: 10.1016/j.cmet.2011.10.010). By expressing mito-roGFP2-Orp1 using a C4da-specific ppk-GAL4 driver, we were able to analyze mitochondrial ROS changes upon paclitaxel treatment in C4da neurons. We observed that the level of mitochondrial ROS was significantly increased in C4da neurons upon paclitaxel treatment.

 We have added these data as Figure 4 and have modified the Results section.

Point 5. Loss of Pink1/parkin cause neurodegeneration, whether pink1 cause C4da neurons death was not tested in this context.

Response: We understand the reviewer’s concern about the toxic effect of PINK1 in C4da neurons. However, we observed no significant change in larval growth upon either overexpression or knockdown of PINK1. In addition, the dendrite structure of C4da neurons was not significantly changed upon knockdown of PINK1. Thus, these results suggest that the growth and survival of C4da neuron was not affected by PINK1 at least in our experimental setting. We have mentioned this point in the Results section.

---

## [Decision Letter · Decision Letter 1]

2 Apr 2020

PONE-D-19-31363R1

PINK1 alleviates thermal hypersensitivity in a paclitaxel-induced Drosophila model of peripheral neuropathy

PLOS ONE

Dear Prof Yun,

Thank you for submitting your manuscript to PLOS ONE. After careful consideration, we feel that it has merit but does not fully meet PLOS ONE’s publication criteria as it currently stands. Therefore, we invite you to submit a revised version of the manuscript that addresses the points raised during the review process.

We would appreciate receiving your revised manuscript by May 17 2020 11:59PM. To enhance the reproducibility of your results, we recommend that if applicable you deposit your laboratory protocols in protocols.io, where a protocol can be assigned its own identifier (DOI) such that it can be cited independently in the future. For instructions see: http://journals.plos.org/plosone/s/submission-guidelines#loc-laboratory-protocols

We look forward to receiving your revised manuscript.

Kind regards,

David Chau

Academic Editor

PLOS ONE

Reviewers' comments:

Reviewer's Responses to Questions

**Comments to the Author**

1. If the authors have adequately addressed your comments raised in a previous round of review and you feel that this manuscript is now acceptable for publication, you may indicate that here to bypass the “Comments to the Author” section, enter your conflict of interest statement in the “Confidential to Editor” section, and submit your "Accept" recommendation.

Reviewer #1: (No Response)

Reviewer #2: All comments have been addressed

2. Is the manuscript technically sound, and do the data support the conclusions?

Reviewer #1: Partly

Reviewer #2: Partly

3. Has the statistical analysis been performed appropriately and rigorously? 

Reviewer #1: Yes

Reviewer #2: Yes

4. Have the authors made all data underlying the findings in their manuscript fully available?

Reviewer #1: Yes

Reviewer #2: Yes

5. Is the manuscript presented in an intelligible fashion and written in standard English?

Reviewer #1: Yes

Reviewer #2: Yes

6. Review Comments to the Author

Reviewer #1: I think this manuscript is almost finished, but a few issues need to be addressed.

1. I think it is essential to show the raw control vs pink1 (overexpression or RNAi) withdrawal responses side by side with significance assessed to highlight if there is a significant baseline difference in the main figures of the manuscript, then show the relative changes from there. The issue is the baseline looks different, and the response to paclitaxel is different, and both messages should be clearly presented in the main figures to help the reader fully understand the results.

2. Figure 2a as presented makes it look like there is no baseline change in nociceptive response to 40C, but figure S1 40C shows what looks like a strong difference at 40C, and I feel this needs to be dealt with upfront so that the data and normalized differences don’t get misinterpreted. Similar issue with Figure 2a vs S2. Then for both the authors should discuss how the baseline difference could confound interpretation of the paclitaxel data so the reader is led toward a more complete understanding of the results.

3. The figures are out of order in the PDF for some reason, also the figure quality is still poor for some reason, please make sure the final published figures are legible, I understand the PDF conversion process can cause this.

4. Figure 1C etc, “relative withdrawal (%)” is sort of confusing, as presented it at first glance seems like paclitaxel reduces sensitivity. It might make more sense to present the data as % sensitization or similar. In general I find “decreased MWL” difficult to conceptualize and I feel “sensitization” is easier for the non-pain expert to understand.

Reviewer #2: The authors have addressed all the concerns I have. It would be great if they had some mechanistic analysis rahter than the descriptive data only.

7. PLOS authors have the option to publish the peer review history of their article (what does this mean?). If published, this will include your full peer review and any attached files.

Reviewer #1: No

Reviewer #2: No

---

## [Author Response · Author response to Decision Letter 1]

1 Jun 2020

May 31, 2020

Dear Editor,

Thank you very much for reviewing our manuscript entitled “PINK1 alleviates thermal hypersensitivity in a paclitaxel-induced Drosophila model of peripheral neuropathy” (PONE-D-19-31363R1). We sincerely appreciate the reviewers’ valuable comments and suggestions. We have modified the manuscript extensively and have included additional data based on the reviewers’ suggestions, which we believe have strengthened the manuscript.

We hope that we have fully addressed the concerns of each reviewer, and that the revised manuscript now meets the standards for publication in PLoS One.

Each concern raised by the reviewers is carefully addressed point by point below.

Reviewer #1: I think this manuscript is almost finished, but a few issues need to be addressed.

1. I think it is essential to show the raw control vs pink1 (overexpression or RNAi) withdrawal responses side by side with significance assessed to highlight if there is a significant baseline difference in the main figures of the manuscript, then show the relative changes from there. The issue is the baseline looks different, and the response to paclitaxel is different, and both messages should be clearly presented in the main figures to help the reader fully understand the results.

Response: We agree with the reviewer that we should show the raw thermal nociceptive response in the main figure before we show the relative withdrawal response. According to the reviewer’s suggestion, we now show the thermal nociceptive responses of control (ppk>w1118) and PINK1-overexpressing (ppk>PINK1) larvae in Fig. 2A. The thermal nociceptive responses of control (ppk>GFP RNAi) and PINK1 RNAi (ppk>PINK1 RNAi) larvae are also shown in Fig. 3A.

In addition, we show the raw heat probe assay results of control (ppk>w1118) and PINK1-overexpressing (ppk>PINK1) larvae in Fig. 2B, and the relative withdrawal changes are shown in Fig. 2C.

We appreciate the reviewer for pointing out this issue. By considering the effects of PINK1 overexpression and knockdown on thermal nociception and the dendrite structure of C4da neurons, we found that PINK1 may alleviate paclitaxel-induced thermal hypersensitivity by means other than preventing alterations in the sensory dendrites of C4da neurons. Thus, we modified the manuscript as we discuss in our next response below.

2. Figure 2a as presented makes it look like there is no baseline change in nociceptive response to 40C, but figure S1 40C shows what looks like a strong difference at 40C, and I feel this needs to be dealt with upfront so that the data and normalized differences don’t get misinterpreted. Similar issue with Figure 2a vs S2. Then for both the authors should discuss how the baseline difference could confound interpretation of the paclitaxel data so the reader is led toward a more complete understanding of the results.

Response: We agree with the reviewer that the effect of PINK1 overexpression should be considered carefully. By analyzing the results upon PINK1 overexpression, we found that PINK1 expression significantly changed thermal nociception as well as the dendrite structure of C4da neurons. Nevertheless, our results suggest that PINK1 expression significantly alleviates paclitaxel-induced thermal hypersensitivity in larvae, and that PINK1 may regulate thermal nociception and dendrite structure through different mechanisms. We believe that the results obtained from PINK1-knockdown experiments further support this notion. Therefore, we have carefully revised the manuscript to clearly state our interpretation of the results. The major changes are shown below.

(Page 9 line 1- Page 10 line 3) 

Ectopic expression of PINK1 rescues the thermal hypersensitivity phenotype induced by paclitaxel treatment

To examine the effect of PINK1 on paclitaxel-induced thermal sensory hypersensitivity, we expressed PINK1 specifically within C4da neurons using the ppk-GAL4 driver [20] and examined thermal nociception upon paclitaxel treatment. We found that PINK1 overexpression significantly reduced thermal sensitivity (Fig. 2A). Larvae expressing PINK1 exhibited approximately 13% to 48 % increased withdrawal latency with different heat probe temperature, suggesting that PINK1 is implicated in thermal nociception. Interestingly, we observed that PINK1 overexpression within C4da neurons significantly suppressed the sensitivity to heat upon paclitaxel treatment. While paclitaxel reduced the MWL by 2.41 sec (from 5.88 sec to 3.47 sec) in control larvae (ppk>w1118), the MWL was decreased by 0.99 sec (from 8.08 sec to 7.09 sec) upon paclitaxel treatment in larvae expressing PINK1 (ppk>PINK1) (Fig. 2B). In terms of the relative changes, paclitaxel reduced the MWL by 42% in control larvae, while the MWL was decreased by only 12% in larvae expressing PINK1 upon paclitaxel treatment (Fig. 2C). These results suggest that PINK1 expression significantly alleviated paclitaxel-induced thermal hypersensitivity in a thermal nociception assay. 

We then assessed the dendritic arborization of C4da neurons, which is known to be associated with the thermal nociception [8]. We observed significant decreases in dendrite length and the number of branch points in larvae expressing PINK1 (Fig. 2D and E). Ectopic expression of PINK1 reduced the dendrite length and the number of branch points of C4da neurons by 51% and 53%, respectively, compared to control larvae, suggesting that the reduced thermal sensitivity upon PINK1 expression may be due to decreased dendritic arborization of C4da neurons. Because PINK1 expression suppressed paclitaxel-induced thermal hypersensitivity, we expected that dendritic arborization would increase less in larvae expressing PINK1 than in control larvae. However, paclitaxel treatment further increased dendritic arborization in larvae expressing PINK1 compared with control larvae (Fig. 2D and E). Quantitative analysis of multiple C4da neurons (n≧ 5) revealed that the dendrite length was increased upon paclitaxel treatment by 32% in control larvae and by 42% in larvae expressing PINK1 (Fig. 2E). The number of dendrite branch points was increased upon paclitaxel treatment by 49% and 70% in control larvae and in larvae expressing PINK1 respectively (Fig. 2E). These results suggest that suppressing alteration of dendrite arborization may not be the only mechanism through which PINK1 alleviates the thermal hypersensitivity induced by paclitaxel treatment.

(Page 10 line 13- Page 11 line 5)

PINK1 knockdown induces thermal hypersensitivity.

To further test the role of PINK1 in thermal nociception, we next examined the effect of specific knockdown of PINK1 expression in C4da neurons. Whereas ectopic PINK1 expression reduced thermal sensitivity in the heat probe assay, C4da neuron-specific knockdown of PINK1 using the ppk-GAL4 driver resulted in significant sensitization of L3 larvae to different heat probe temperatures in the thermal nociception assay (Fig. 3A). PINK1 knockdown decreased the withdrawal latency to the 40 ºC heat probe to 41% of the control level (Fig. 3B), confirming that PINK1 plays an important role in thermal nociception. 

To understand the role of PINK1 in the sensory dendrites of C4da neurons, we also examined the dendritic structure of C4da neurons upon C4da neuron-specific knockdown of PINK1. Interestingly, we observed no change in the dendritic structure of C4da neurons upon PINK1 knockdown (Fig. 3C). The quantitative analysis of multiple C4da neurons (n≧ 5) also showed that both dendrite length and the number of dendrite branches (Fig. 3D) were not significantly changed upon PINK1 knockdown. These results indicate that knockdown of PINK1 in C4 da neurons induces an increase in thermal nociception without changes in the dendritic structure. We also found that the size of larvae was not significantly changed upon PINK1 knockdown (Fig. 3E), confirming that PINK1 has no effect on larval growth. These results suggest that PINK1 may separately regulate thermal nociception and dendrite structure through independent mechanisms.

3. The figures are out of order in the PDF for some reason, also the figure quality is still poor for some reason, please make sure the final published figures are legible, I understand the PDF conversion process can cause this.

Response: We apologize for this error. We carefully checked the order of the figures this revision. 

Regarding the quality of the figures, we have prepared all the original images carefully according to the figure preparation guidelines of PLoS One. All of the submitted figure images are of 300 dpi resolution. Thus, we believe that the quality of original figures will meet the standard of the journal.

4. Figure 1C etc, “relative withdrawal (%)” is sort of confusing, as presented it at first glance seems like paclitaxel reduces sensitivity. It might make more sense to present the data as % sensitization or similar. In general I find “decreased MWL” difficult to conceptualize and I feel “sensitization” is easier for the non-pain expert to understand.

Response: To address the reviewer’s concern about the term “relative withdrawal (%)”, we instead use the term “withdrawal latency (% baseline)”, which is commonly used in pain experiments, in Fig. 2C. The results in Fig. 1C are now expressed as “withdrawal latency” instead of the relative values.

To facilitate readers’ understanding, we have reduced the use of the term “mean withdrawal latency (MWL)” and revised the manuscript by using the term “thermal sensitivity” or “sensitization” as shown below.

(Page 9, line 5) 

We found that PINK1 overexpression significantly reduced thermal sensitivity (Fig. 2A). Larvae expressing PINK1 exhibited approximately 13% to 48% increased withdrawal latency with different heat probe temperatures,…

(Page 10, line 15) 

Whereas ectopic PINK1 expression reduced thermal sensitivity in the heat probe assay, C4da neuron-specific knockdown of PINK1 using the ppk-GAL4 driver resulted in significant sensitization of L3 larvae to different heat probe temperatures in the thermal nociception assay (Fig. 3A). PINK1 knockdown decreased the withdrawal latency to the 40 ºC heat probe to 41% of the control level (Fig. 3B),…

Reviewer #2: The authors have addressed all the concerns I have. It would be great if they had some mechanistic analysis rahter than the descriptive data only.

Response: We agree with the reviewer that understanding the precise mechanism through which PINK1 expression suppresses paclitaxel-induced thermal hypersensitivity is important. In particular, how PINK1 reduces paclitaxel-induced mitochondrial dysfunction in sensory neurons needs to be explored in further studies. We are currently conducting experiments to understand the molecular mechanisms and hope that we are able to report the results soon.

---

## [Decision Letter · Decision Letter 2]

24 Jun 2020

PONE-D-19-31363R2

PINK1 alleviates thermal hypersensitivity in a paclitaxel-induced Drosophila model of peripheral neuropathy

PLOS ONE

Dear Dr. Yun,

Thank you for submitting your manuscript to PLOS ONE. After careful consideration, we feel that it has merit but does not fully meet PLOS ONE’s publication criteria as it currently stands. Therefore, we invite you to submit a revised version of the manuscript that addresses the points raised during the review process.

We look forward to receiving your revised manuscript.

Kind regards,

David Chau

Academic Editor

PLOS ONE

Reviewers' comments:

Reviewer's Responses to Questions

**Comments to the Author**

1. If the authors have adequately addressed your comments raised in a previous round of review and you feel that this manuscript is now acceptable for publication, you may indicate that here to bypass the “Comments to the Author” section, enter your conflict of interest statement in the “Confidential to Editor” section, and submit your "Accept" recommendation.

Reviewer #1: All comments have been addressed

Reviewer #2: All comments have been addressed

2. Is the manuscript technically sound, and do the data support the conclusions?

Reviewer #1: Yes

Reviewer #2: Partly

3. Has the statistical analysis been performed appropriately and rigorously? 

Reviewer #1: Yes

Reviewer #2: No

4. Have the authors made all data underlying the findings in their manuscript fully available?

Reviewer #1: Yes

Reviewer #2: Yes

5. Is the manuscript presented in an intelligible fashion and written in standard English?

Reviewer #1: Yes

Reviewer #2: Yes

6. Review Comments to the Author

Reviewer #1: The authors have addressed all of my concerns and I believe the manuscript is suitable for publicaiton.

Reviewer #2: The current manuscript is significantly improved, however, it is overall descriptive, mechanistic analysis is largely missing.

In Figure 5, the authors observed that paclitaxel induced mitophagy was suppressed by pink1 OE using mitoKeima reporter. The authors then claimed that PINK1 reduced paclitaxel-induced increases in mitophagy levels.

Whether pink1 suppress or accelerate mitophagy in this context needs to be clarified, since previously reports showed that pink1/parkin KD muscles or DA neurons have less mitophagy activity in Drosophila (eLife 2018;7:e35878 doi: 10.7554/eLife.35878).

The authors can easily address this by manipulating downstream events of mitophagy, such as lysosome and proteasome activity in pink1 LOF background.

7. PLOS authors have the option to publish the peer review history of their article (what does this mean?). If published, this will include your full peer review and any attached files.

Reviewer #1: No

Reviewer #2: No

---

## [Author Response · Author response to Decision Letter 2]

6 Aug 2020

August 7, 2020

Dear Editor,

Thank you very much for reviewing our manuscript entitled “PINK1 alleviates thermal hypersensitivity in a paclitaxel-induced Drosophila model of peripheral neuropathy” (PONE-D-19-31363R2). We sincerely appreciate the reviewers’ valuable comments and suggestions. We have modified the manuscript and have included additional data based on the reviewers’ suggestions, which we believe have strengthened the manuscript.

We hope that we have fully addressed the concerns of each reviewer, and that the revised manuscript now meets the standards for publication in PLoS One.

Each concern raised by the reviewers is carefully addressed point by point below.

Reviewer #1: The authors have addressed all of my concerns and I believe the manuscript is suitable for publication.

Reviewer #2: The current manuscript is significantly improved, however, it is overall descriptive, mechanistic analysis is largely missing.

In Figure 5, the authors observed that paclitaxel induced mitophagy was suppressed by pink1 OE using mitoKeima reporter. The authors then claimed that PINK1 reduced paclitaxel-induced increases in mitophagy levels.

Whether pink1 suppress or accelerate mitophagy in this context needs to be clarified, since previously reports showed that pink1/parkin KD muscles or DA neurons have less mitophagy activity in Drosophila (eLife 2018;7:e35878 doi: 10.7554/eLife.35878).

The authors can easily address this by manipulating downstream events of mitophagy, such as lysosome and proteasome activity in pink1 LOF background.

Response: We understand the reviewer’s concern about the mitophagy change upon PINK1 overexpression. We repeated the Fig. 5 experiment and mitophagy analysis with larger number of samples and found that the basal mitophagy level of the C4da neuron was not significantly changed upon PINK1 expression. As the reviewer mentioned, Wim Vandenberghe’s group has shown that loss of PINK1 and knockdown of parkin abolish the age-dependent increase of mitophagy in muscle while basal mitophagy is unchanged (Cornelissen T et al, 2018, eLife, 7:e35878). We have also previously shown that the mitophagy level of DA neurons was not changed by knockdown of PINK1, but mitophagy inductions upon hypoxia, and rotenone treatment were abolished (Kim YY et al, 2019, FASEB J, 33:9742-9751. doi: 10.1096/fj.201900073R), suggesting that PINK1 function is dispensable for basal mitophagy. We also observed that the mitophagy level of C4da neurons was not changed by knockdown of PINK1. Given the significant increase in thermal sensitivity upon PINK1 knockdown, the unchanged mitophagy activity suggested that PINK1 may restore mitochondrial homeostasis through mechanisms other than mitophagy. Previous studies have indicated that PINK1 controls mitochondrial quality through additional methods such as regulating complex I activity and mitochondrial transport (Reviewed in Voigt A et al, 2016, J Neurochem, 139:232-239, doi: 10.1111/jnc.13655; Steer EK et al, 2015, Antioxid Redox Signal, 22(12):1047-1059 doi: 10.1089/ars.2014.6206). A body of studies even suggest that PINK1 suppresses mitophagy activity in certain cellular contexts through a direct or indirect manner (Reviewed in Steer EK et al, 2015, Antioxid Redox Signal, 22(12):1047-1059 doi: 10.1089/ars.2014.6206). Therefore, the exact mechanism by which PINK1 ameliorates paclitaxel-induced mitochondrial dysfunction in C4da neuron should be carefully determined in further studies.

We have revised the discussion to clearly state these points.

We replaced Fig. 5B with a new results and added the mitophagy results upon PINK1 knockdown as Supplementary FigS1.

---

## [Decision Letter · Decision Letter 3]

26 Aug 2020

PONE-D-19-31363R3

PINK1 alleviates thermal hypersensitivity in a paclitaxel-induced Drosophila model of peripheral neuropathy

PLOS ONE

Dear Dr. Yun,

Thank you for submitting your manuscript to PLOS ONE. After careful consideration, we feel that it has merit but does not fully meet PLOS ONE’s publication criteria as it currently stands. Therefore, we invite you to submit a revised version of the manuscript that addresses the points raised during the review process.

We look forward to receiving your revised manuscript.

Kind regards,

David Chau

Academic Editor

PLOS ONE

Reviewers' comments:

Reviewer's Responses to Questions

**Comments to the Author**

1. If the authors have adequately addressed your comments raised in a previous round of review and you feel that this manuscript is now acceptable for publication, you may indicate that here to bypass the “Comments to the Author” section, enter your conflict of interest statement in the “Confidential to Editor” section, and submit your "Accept" recommendation.

Reviewer #2: All comments have been addressed

Reviewer #3: (No Response)

2. Is the manuscript technically sound, and do the data support the conclusions?

Reviewer #2: No

Reviewer #3: Partly

3. Has the statistical analysis been performed appropriately and rigorously? 

Reviewer #2: No

Reviewer #3: Yes

4. Have the authors made all data underlying the findings in their manuscript fully available?

Reviewer #2: Yes

Reviewer #3: Yes

5. Is the manuscript presented in an intelligible fashion and written in standard English?

Reviewer #2: Yes

Reviewer #3: Yes

6. Review Comments to the Author

Reviewer #2: In this version, the authors have addressed all my concerns, and is significantly improved. As the authors discussed, the underlying mechanism need to be adressed in the future.

Reviewer #3: In the present well written study the researchers aim to shed light on the in vivo mechanism of chemotherapy-induced peripheral neuropathy (CIPN), a common side-effect of chemotherapy treatment, and induced by the anticancer drug paclitaxel. They use a recently published drosophila model of paclitaxel-induced thermal hyperalgesia in L3 larvae. Whilst some aspects of paclitaxel mechanism of action are known, how it induces peripheral neuropathy is less well understood. Here, the authors examined the role of PINK1, a protein implicated in mitochondrial homeostasis and quality control in paclitaxel-induced thermal hyperalgesia.

The authors' conclusion that PINK1 plays a role in thermal sensitivity is supported by their data showing that ectopic expression of PINK1 in class IV dendritic aborization sensory neurons under basal conditions led to increased withdrawal latency times to the heat stimulus compared with background controls. Moreover, KD of PINK1 induced a reduced withdrawal latency compared with a control RNAi genotype.

PINK1 was further implicated in thermal hyperalgesia as ectopic expression of PINK1 (as above), dampened the ability of paclitaxel to induce increased withdrawal latency times compared to vehicle control. Thus, the authors’ conclusion that PINK1 has a neuroprotective function in a paclitaxel-induced peripheral neuropathy model is substantiated by the data.

The authors’ conclusion that the neuroprotective role of PINK in paclitaxel-induced thermal hyperalgesia is via mechanisms independent of dendritic organisation is supported by their findings that PINK1 expression reduced measures of dendritic arborization (dendritic branching and length) but did not suppress paclitaxel-induced increases, whilst PINK1 KD did not change these parameters.

The authors attempted to address the underlying mechanism of this neuroprotective effect by examining aspects of mitochondrial dysfunction. They found that paclitaxel treatment induced mitochondrial reactive oxygen species production and mitophagy, using established in vivo tools, and that PINK1 expression ameliorated the induction of mitophagy by paclitaxel. Additionally, basal levels of mitophagy were not changed by either ectopic expression or knockdown of PINK1. The authors concluded that PINK1 modulates thermal sensitivity by regulating mitochondrial homeostasis, by mechanisms independently of mitophagy. Whilst the findings can be supported by appropriate citations, they are observational in manner and further mechanistic insight was not gained. The authors acknowledge that further studies are necessary to understand the current findings within the context of the in vivo work within the field.

The work presented in this manuscript investigates an important area in the field by using established reagents and methodologies, thus generating robust results. Although the data which sought to uncover a mitochondrial homeostasis mechanism is mainly observational and somewhat inconclusive, the overall conclusion of the manuscript that PINK1 plays a role in thermal sensitivity is clear. The findings support the conclusion that PINK1 may be a bonafide therapeutic target for chemotherapy-induced peripheral neuropathy worthy of future investigation. Since further work to uncover mechanistic insights is beyond the scope of the manuscript, if the authors address the points below it would be suitable for publication in PLOS ONE.

Major point:

1. The observation that ectopic expression of PINK1 ameliorates paclitaxel-induced thermal hyperalgesia and mitophagy would be more compelling if a benign UAS control gene was used to account for the use of the GAL4-UAS system for expression of PINK. However, this is not necessarily a requirement for publication since there are conflicting conventions in the field regarding the use of such controls. Please can the authors justify the controls used in Figure 2 and Figure 5.

Minor points:

1. Consistency with GAL4 notation on page 4 Material and Methods section Drosophila strains.

2. Figure 1D, Figure 2D and Figure 3C - please add a scale bar to the representative images and detail in the legends.

3. Please add in the citation Lee et al, J Cell Biol. 2018 May 7; 217(5): 1613–1622. doi: 10.1083/jcb.201801044 to page 14 in relevance to loss of PINK1 minimally affecting basal mitophagy in Drosophila

7. PLOS authors have the option to publish the peer review history of their article (what does this mean?). If published, this will include your full peer review and any attached files.

Reviewer #2: No

Reviewer #3: No

---

## [Author Response · Author response to Decision Letter 3]

28 Aug 2020

August 28, 2020

Dear Editor,

Thank you very much for reviewing our manuscript entitled “PINK1 alleviates thermal hypersensitivity in a paclitaxel-induced Drosophila model of peripheral neuropathy” (PONE-D-19-31363R3). We sincerely appreciate the reviewers’ valuable comments and suggestions. We have modified the manuscript and have included additional data based on the reviewers’ suggestions, which we believe have strengthened the manuscript.

We hope that we have fully addressed the concerns of each reviewer, and that the revised manuscript now meets the standards for publication in PLoS One.

Each concern raised by the reviewers is carefully addressed point by point below.

Reviewer #2: In this version, the authors have addressed all my concerns, and is significantly improved. As the authors discussed, the underlying mechanism need to be adressed in the future.

Reviewer #3: In the present well written study the researchers aim to shed light on the in vivo mechanism of chemotherapy-induced peripheral neuropathy (CIPN), a common side-effect of chemotherapy treatment, and induced by the anticancer drug paclitaxel. They use a recently published drosophila model of paclitaxel-induced thermal hyperalgesia in L3 larvae. Whilst some aspects of paclitaxel mechanism of action are known, how it induces peripheral neuropathy is less well understood. Here, the authors examined the role of PINK1, a protein implicated in mitochondrial homeostasis and quality control in paclitaxel-induced thermal hyperalgesia.

The authors' conclusion that PINK1 plays a role in thermal sensitivity is supported by their data showing that ectopic expression of PINK1 in class IV dendritic aborization sensory neurons under basal conditions led to increased withdrawal latency times to the heat stimulus compared with background controls. Moreover, KD of PINK1 induced a reduced withdrawal latency compared with a control RNAi genotype.

PINK1 was further implicated in thermal hyperalgesia as ectopic expression of PINK1 (as above), dampened the ability of paclitaxel to induce increased withdrawal latency times compared to vehicle control. Thus, the authors’ conclusion that PINK1 has a neuroprotective function in a paclitaxel-induced peripheral neuropathy model is substantiated by the data.

The authors’ conclusion that the neuroprotective role of PINK in paclitaxel-induced thermal hyperalgesia is via mechanisms independent of dendritic organisation is supported by their findings that PINK1 expression reduced measures of dendritic arborization (dendritic branching and length) but did not suppress paclitaxel-induced increases, whilst PINK1 KD did not change these parameters.

The authors attempted to address the underlying mechanism of this neuroprotective effect by examining aspects of mitochondrial dysfunction. They found that paclitaxel treatment induced mitochondrial reactive oxygen species production and mitophagy, using established in vivo tools, and that PINK1 expression ameliorated the induction of mitophagy by paclitaxel. Additionally, basal levels of mitophagy were not changed by either ectopic expression or knockdown of PINK1. The authors concluded that PINK1 modulates thermal sensitivity by regulating mitochondrial homeostasis, by mechanisms independently of mitophagy. Whilst the findings can be supported by appropriate citations, they are observational in manner and further mechanistic insight was not gained. The authors acknowledge that further studies are necessary to understand the current findings within the context of the in vivo work within the field.

The work presented in this manuscript investigates an important area in the field by using established reagents and methodologies, thus generating robust results. Although the data which sought to uncover a mitochondrial homeostasis mechanism is mainly observational and somewhat inconclusive, the overall conclusion of the manuscript that PINK1 plays a role in thermal sensitivity is clear. The findings support the conclusion that PINK1 may be a bonafide therapeutic target for chemotherapy-induced peripheral neuropathy worthy of future investigation. Since further work to uncover mechanistic insights is beyond the scope of the manuscript, if the authors address the points below it would be suitable for publication in PLOS ONE.

Major point:

1. The observation that ectopic expression of PINK1 ameliorates paclitaxel-induced thermal hyperalgesia and mitophagy would be more compelling if a benign UAS control gene was used to account for the use of the GAL4-UAS system for expression of PINK. However, this is not necessarily a requirement for publication since there are conflicting conventions in the field regarding the use of such controls. Please can the authors justify the controls used in Figure 2 and Figure 5.

Response: We understand the reviewer’s concern about the control for PINK1 expression. As stated in the beginning of the result section (page 8), we adopted a Drosophila thermal nociception assay from recently performed studies. In particular, experiments investigating the effect of PINK1 expression were conducted in accordance with the experimental setting of Grace Zhai group’s recent paper (Grazil J et al. 2018, Dis Model Mech, DOI: 10.1242/dmm.032938). We used a ppk-GAL4 fly as a control for the ppk-GAL4-drived expression of PINK1, because Grazil J et al also used ppk-GAL4 as a control when they examined the effect of Nmnat expression on paclitaxel-induced hyperalgesia in their study. 

We have revised the result and figure legends to state more clearly the control for PINK1 expression.

(Page 9 line 2) To examine the effect of PINK1 on paclitaxel-induced thermal sensory hypersensitivity, we expressed PINK1 specifically within C4da neurons using the ppk-GAL4 driver [20] and examined thermal nociception upon paclitaxel treatment using a ppk-GAL4 as a control.

(Page 17 line 18) Figure 2. PINK1 mitigates the heat-hyperalgesia phenotype induced by paclitaxel treatment.

(A) The thermal nociceptive response of ppk-GAL4 control L3 larvae (ppk>w1118) and larvae expressing PINK1 in C4da sensory neuron (ppk>PINK1) to heat probes at different temperatures.

Minor points:

1. Consistency with GAL4 notation on page 4 Material and Methods section Drosophila strains.

Response: We thank the reviewer for correct this mistake. We changed the “ppk-Gal4” to “ppk-GAL4” for the consistent description.

2. Figure 1D, Figure 2D and Figure 3C - please add a scale bar to the representative images and detail in the legends.

Response: Thank you again for this comment. We added scale bars at the pictures in Figure 1D, Figure 2D and Figure 3C. We have revised Figure legends accordingly.

3. Please add in the citation Lee et al, J Cell Biol. 2018 May 7; 217(5): 1613–1622. doi: 10.1083/jcb.201801044 to page 14 in relevance to loss of PINK1 minimally affecting basal mitophagy in Drosophila

Response: According to the reviewer’s suggestion, we have added the recommended paper to the discussion section.

---

## [Decision Letter · Decision Letter 4]

1 Sep 2020

PINK1 alleviates thermal hypersensitivity in a paclitaxel-induced Drosophila model of peripheral neuropathy

PONE-D-19-31363R4

Dear Dr. Yun,

We’re pleased to inform you that your manuscript has been judged scientifically suitable for publication and will be formally accepted for publication once it meets all outstanding technical requirements.

Kind regards,

David Chau

Academic Editor

PLOS ONE

Additional Editor Comments (optional):

Reviewers' comments:

Reviewer's Responses to Questions

**Comments to the Author**

1. If the authors have adequately addressed your comments raised in a previous round of review and you feel that this manuscript is now acceptable for publication, you may indicate that here to bypass the “Comments to the Author” section, enter your conflict of interest statement in the “Confidential to Editor” section, and submit your "Accept" recommendation.

Reviewer #3: All comments have been addressed

2. Is the manuscript technically sound, and do the data support the conclusions?

Reviewer #3: Yes

3. Has the statistical analysis been performed appropriately and rigorously? 

Reviewer #3: Yes

4. Have the authors made all data underlying the findings in their manuscript fully available?

Reviewer #3: Yes

5. Is the manuscript presented in an intelligible fashion and written in standard English?

Reviewer #3: Yes

6. Review Comments to the Author

Reviewer #3: In the revised manuscript the authors have satisfactorily address the concerns and comments raised.

7. PLOS authors have the option to publish the peer review history of their article (what does this mean?). If published, this will include your full peer review and any attached files.

Reviewer #3: No